# High-Strength, High-Water-Retention Hemicellulose-Based Hydrogel and Its Application in Urea Slow Release

**DOI:** 10.3390/ijms24119208

**Published:** 2023-05-24

**Authors:** Yajun Hou, Baojuan Deng, Shanshan Wang, Yun Ma, Xing Long, Fei Wang, Chengrong Qin, Chen Liang, Shuangquan Yao

**Affiliations:** Guangxi Key Laboratory of Clean Pulp & Papermaking and Pollution Control, School of Light Industrial and Food Engineering, Guangxi University, Nanning 530004, China; 2016301008@st.gxu.edu.cn (Y.H.); dengbaojuan@st.gxu.edu.cn (B.D.); 2116391037@st.gxu.edu.cn (S.W.); 1916401003@st.gxu.edu.cn (Y.M.); 2216301033@st.gxu.edu.cn (X.L.); 2116391035@st.gxu.edu.cn (F.W.); qinchengrong@gxu.edu.cn (C.Q.); liangchen@st.gxu.edu.cn (C.L.)

**Keywords:** antioxidant, hemicellulose-based hydrogels, urea sustained release, water retention performance

## Abstract

The use of fertilizer is closely related to crop growth and environmental protection in agricultural production. It is of great significance to develop environmentally friendly and biodegradable bio-based slow-release fertilizers. In this work, porous hemicellulose-based hydrogels were created, which had excellent mechanical properties, water retention properties (the water retention ratio in soil was 93.8% after 5 d), antioxidant properties (76.76%), and UV resistance (92.2%). This improves the efficiency and potential of its application in soil. In addition, electrostatic interaction and coating with sodium alginate produced a stable core–shell structure. The slow release of urea was realized. The cumulative release ratio of urea after 12 h was 27.42% and 11.38%, and the release kinetic constants were 0.0973 and 0.0288, in aqueous solution and soil, respectively. The sustained release results demonstrated that urea diffusion in aqueous solution followed the Korsmeyer–Peppas model, indicating the Fick diffusion mechanism, whereas diffusion in soil adhered to the Higuchi model. The outcomes show that urea release ratio may be successfully slowed down by hemicellulose hydrogels with high water retention ability. This provides a new method for the application of lignocellulosic biomass in agricultural slow-release fertilizer.

## 1. Introduction

The world’s population increased from 6.1 billion to 7.8 billion between 2000 and 2020. However, the arable land decreased by 75 million hectares according to the 2020 Statistical Year-book of World Food and Agriculture. Mankind strives to produce food as efficiently as possible [1]. In fact, fertilizer application plays a crucial role in improving crop yield and quality [2]. Crop growth is aided by the essential ingredient of nitrogen [3]. Urea, due to its high nitrogen content and relatively low cost, is mainly used in agricultural fertilizers [4]. However, raw urea is used directly in conventional agricultural production [3]. India was the world’s largest importer of urea in 2020, accounting for 20.4%. This has led to increased overuse of urea, which loses effectiveness with time [5]. More dangerously, it creates soil pollution, which will undermine the healthy agricultural ecological balance [2,6]. In addition, water is essential for crop growth, which can promote the efficiency of urea [7]. This indicates that the development of slow-release urea systems with water-retention capabilities has significant commercial and environmental benefit [8,9]. Continuous nitrogen fertilizer and moisture were provided using a slow-release system. Urea loss and frequency of application were reduced compared with conventional direct fertilization. With an increase in urea efficiency, excessive fertilization-related environmental contamination is prevented.

The use of hydrogels as fertilizer carriers has received a great deal of attention recently [10,11,12]. Hydrogels are hydrophilic three-dimensional polymers [13]. They possess a high-crosslinking network structure, which gives them excellent water absorption and storage capacity [4]. The combination of hydrogels with slow-release fertilizers is advantageous as it helps to slow down the release ratio of nutrients while maintaining water content [14]. Furthermore, it is necessary to develop biodegradable materials. The term “biodegradability” describes the potential for microorganisms to break down environmental contaminants [15,16]. The biodegradability of hydrogels is enhanced by the addition of biodegradable natural biomass. Dependence on petroleum-based monomers is reduced [7]. Hemicellulose (HC) is mainly composed of five-carbon sugars and six-carbon sugars and is an important component of bio-based polymers [17,18,19,20]. It includes xylose, galactose, arabinose, and mannose [21]. HC is characterized by a high degree of branching and a low degree of polymerization. It has gained widespread attention due to its environmental friendliness, good biocompatibility, strong renewability and easy degradation [22]. It is rich in oxygen-containing groups [23], such as hydroxyl, acetyl, and carboxyl groups. It can be modified by etherification [24], esterification [25], and grafting copolymerization [26]. As a result, hemicellulose-based hydrogels are versatile [27,28].

Unfortunately, although HC-based hydrogels have a wide range of applications and rich functional groups, their lack of mechanical strength seriously reduces their usability [29,30,31,32]. The improvement of mechanical properties of HC-based hydrogels has attracted extensive attention. Kong et al. [33] prepared HC-based composite hydrogels via free radical polymerization and coordination of carboxyl groups and Al^3+^. The mechanical properties of hydrogels can be significantly enhanced by the coordination action of Al^3+^. Essawy et al. [34] prepared hydrogels with better mechanical properties by grafting acrylic acid (AA) and cellulose with chitosan (CSN) and chemical bonding. Wang et al. [35] prepared hydrogels using a mixture of k-carrageenan and sodium alginate (SA), which presented excellent mechanical qualities. Furthermore, the addition of mixed coating improved water retention and slow-release ability. It was found that the mechanical properties of hydrogels could be improved by adding natural polymer materials [36]. Other natural polymers added to urea’s slow release have an unsatisfactory impact, however. There are few reports on improving the water retention performance of HC hydrogels and their application in urea slow release.

In this work, AA was grafted onto the HC molecular chain using free radicals, and introduced into a CSN-ferric chloride (Fe^3+^) system. A novel bio-based hydrogel (HC-CSN-Fe^3+^) with an envelope structure was prepared by coating sodium alginate for sustained urea release. The prepared hydrogels were characterized by scanning electron microscopy (SEM), Fourier transform infrared spectroscopy (FT-IR), and X-ray photoelectron spectroscopy (XPS) to study the chemical and physical structure of hydrogels. The mechanical properties, oxidation resistance, and UV resistance of hydrogels were investigated. The release ratio of urea after HC-CSN-Fe^3+^ hydrogel coated with sodium alginate was studied. This work fills a gap in the application of HC for urea slow-release materials. HC-based hydrogels have great application potential in the field of urea sustained release and are expected to be used as a good biomass-based slow-release fertilizer.

## 2. Results and Discussion

### 2.1. Microstructure Analysis of HC-CSN-Fe^3+^ Hydrogels

Figure 1 displays the SEM images of the internal pore size structures of various hydrogel samples. The void structure of the hydrogel is improved by the introduction of CSN and Fe^3+^ (Figure 1c). Because of the large and low-density internal pore structure brought on by a single chemical crosslinking, HC-PAA hydrogel has bigger pores. The pore size of the hydrogel with CSN was reduced. This was attributed to the chemical crosslinking between CSN and AA. One of the primary reasons was also the creation of hydroxyl groups between hemicellulose, CSN, and polyacrylic acid. The pore structure decreases, and the number increases, due to the complexation of Fe^3+^ with the CSN amino group in HC-CSN-Fe^3+^ hydrogel. Through enhanced cross-linking, the circulation channels of water molecules are increased [37].

### 2.2. Functional Group Analysis of HC-CSN-Fe^3+^ Hydrogels

The absorption peak of hemicellulose -OH was 3406 cm^−1^ in FT-IR. The telescopic vibration of HC alkanes was 2918 cm^−1^. The typical absorption peak of HC was 1631 cm^−1^. The absorption peak at 899 cm^−1^ was attributed to the β-glycosidic bond [21,22,38]. C-H stretching and C=O tensile vibrations were attributed to CSN at 2859 cm^−1^ and 1659 cm^−1^, respectively [39]. This indicates that CSN was successfully introduced into the HC-CSN and HC-CSN-Fe^3+^ hydrogels. Meanwhile, a weakening of peak intensity at 2918 cm^−1^ was found in three different hydrogel infrared spectra. The results indicated that hydrogen bonding was formed between HC and polyacrylic acid. In addition, the weakening of the peak intensity at 3406 cm^−1^ was ascribed to the deformation of -OH in the free radical reaction [40]. This demonstrated that AA was successfully grafted onto the HC molecular chains. The absorption peak of 650 cm^−1^ in the infrared spectrum of HC-CSN-Fe^3+^ hydrogel was assigned to the vibration of Fe^3+^-O. The outcomes suggested the formation of metal bonds, and that Fe^3+^ was successfully introduced into the HC hydrogel system.

### 2.3. Surface Elemental Analysis of HC-CSN-Fe^3+^ Hydrogels

XPS of HC-PAA hydrogel, HC-CSN hydrogel, and HC-CSN-Fe^3+^ hydrogel are shown in Figure 2a–c, respectively. The peaks at 248.8 eV, 398.1 eV, 531.8 eV, and 713.1 eV represent C1s, N1s, O1s, and Fe 2p, respectively. The C1s peak can be divided into three peaks at 248.8 eV, 285.8 eV, and 288.9 eV, which represent the C-C bond, C-N/C-O bond, and C=O bond, respectively [41]. The N1s peak can be divided into two peaks at 398.1 eV and 400.3 eV, which represent the -NH_2_ and -NH-acetyl groups, respectively. The presence of the Fe 2p peak in HC-CSN-Fe^3+^ hydrogel indicates the successful incorporation of Fe^3+^ into the hydrogel network. The C-C peak of HC-CSN hydrogel and HC-CSN-Fe^3+^ hydrogel was significantly enhanced with the addition of CSN. This indicated an increase in carbon chain length due to the successful connection of CSN with HC and the introduction of carbon-containing groups. Furthermore, the appearance of an -NH-acetyl peak in HC-CSN hydrogel and HC-CSN-Fe^3+^ hydrogel verified the occurrence of a dehydration reaction. The presence of the -NH_2_ peak suggested that some CSN did not participate in the dehydration reaction. The enhanced -NH_2_ peak in HC-CSN-Fe^3+^ hydrogel was attributed to the chelation of some Fe^3+^ with -NH_2_ groups on CSN, which prevents them from participating in the dehydration reaction.

### 2.4. Effects of Hemicellulose and Chitosan Content on HC-CSN-Fe^3+^ Hydrogel Properties

In actuality, the performance of hydrogels is greatly influenced by the addition amounts of HC and CSN, Fe^3+^ concentration, and AA addition. Therefore, we designed a single-factor experiment to explore the effects of different contents on hydrogels, as is shown in Table 1. The HC/CSN ratio ranged from 5:1 to 5:5 (g·g^−1^), the concentration of Fe^3+^ ranged from 0.01 to 0.04 (mol·L^−1^), and the addition of AA ranged from 3 to 5 (mL). Free radical copolymerization of HC with AA was performed due to its high active hydroxyl content. The properties of hydrogels were changed due to HC content affecting graft reaction efficiency. CSN dissolves in acidic solutions and forms positively charged cationic groups due to its abundance of amino and hydroxyl groups. It creates a stable network structure by chelating with metal ions. Therefore, the ratios of HC to CSN were studied for their effects on the mechanical properties (strain range of 0–60%) and swelling performance of hydrogels. The ratios studied were 5:1, 5:2, 5:3, 5:4, and 5:5, as shown in Figure 3a,b.

The mechanical properties of hydrogels were enhanced with the introduction of CSN compared with traditional HC hydrogels. The compressive stress of HC-CSN hydrogel increased from 0.008 MPa to 0.012 MPa at 60% of the maximum compressive strain in Figure 3a. This is attributed to the hydrogen bonding between the abundant hydroxyl groups of CSN and HC. Furthermore, the mechanical properties of hydrogels were enhanced due to the graft reaction of CSN with polyacrylic acid. The mechanical properties of the hydrogel were significantly improved after the addition of Fe^3+^. It was found that the compressive stress of the hydrogel reached a maximum at an HC/CSN ratio of 5:3, with a value of 0.036 MPa, which is 350% higher than that of the HC-PAA hydrogel. Because of the increased CSN, there were more ion-binding sites, more amino groups were used to chelate Fe^3+^, and there was better ion bond crosslinking. Additionally, the creation of a dual-network structure as a result of the covalent and ion bond crosslinking considerably enhanced the hydrogel’s mechanical properties [29]. The viscosity of the pre-polymer solution increased with CSN content. An excessive amount of CSN lead to uneven distribution of Fe^3+^, affecting the stability of the physical filling. The cross-linking network of polyacrylic acid was affected, resulting in decreased mechanical properties. The swelling performance, as the most representative property of hydrogels, is influenced by factors such as cross-linking density and hydrophilic groups. The swelling performance of different hydrogels was investigated as shown in Figure 3c. The hydrogels with added Fe^3+^ exhibited significantly increased cross-linking density, resulting in reduced hydrogel porosity and decreased capacity for accommodating water molecules. In addition, the HC-CSN-Fe^3+^ hydrogel reached swelling equilibrium at 16 h, while the HC-PAA hydrogel and HC-CSN hydrogel reached swelling equilibrium after 72 h. This can be attributed to the fact that when the hydrogel has a higher degree of cross-linking, the free water space for accommodation decreases, resulting in a shorter equilibrium swelling time. Consequently, 5:3 is the ideal HC/CSN ratio.

### 2.5. Effect of AA Addition on HC-CSN-Fe^3+^ Hydrogel Properties

As a free radical copolymer, AA is chemically cross-linked with HC and CSN. The degree of chemical crosslinking has a great influence on the properties of hydrogels. Therefore, the effect of AA addition on the performance of the hydrogels was explored, which was 3, 4, and 5 mL, respectively. The results are shown in Figure 3c,d.

The characteristics of hydrogels were dramatically influenced by the amount of AA applied. When the addition amount was less than 3 mL, there was insufficient AA for the free radical copolymerization reaction, which prevented the hydrogel system from forming. Chemical crosslinking is difficult to achieve. The compressive stress increased significantly from 0.012 MPa to 0.038 MPa when the amount of AA increased from 3 mL to 5 mL. At 3 mL AA, it underwent chemical crosslinking with HC and CSN. However, the free radical copolymerization reaction was incomplete. The low degree of grafting resulted in low compression stress of the hydrogel. An excess of AA occurred in the self-polymerization with the amount of AA added exceeding 4 mL, which affected the mechanical properties and swelling properties of the hydrogel. The swelling property of the hydrogel gradually decreased with the increase of the amount of acrylic acid added. This was attributed to the increase of the chemical crosslinking density of the hydrogel. Therefore, the swelling ratio reached the maximum at 3 mL AA. It decreased when the swelling time exceeded 16 h. This was because some HCS dissolve in water without engaging in crosslinking. The swelling ratio at 4 mL AA addition was higher than that at 5 mL AA addition. Therefore, the optimal amount of AA addition was 4 mL.

### 2.6. Effect of Iron Ion Concentration on HC-CSN-Fe^3+^ Hydrogel Properties

The concentration of Fe^3+^ is an important factor affecting the degree of crosslinking. Therefore, the effects of different Fe^3+^ on the performance of hydrogels were studied with amounts of 0.01, 0.02, 0.03, and 0.04 M. The results are shown in Figure 3e,f.

The compressive stress of the hydrogel gradually increased with the increase of Fe^3+^ concentration. This was explained by increased ionic bond creation, higher ionic bond crosslinking, and increased chelation of Fe^3+^ with CSN [33]. The compressive stress of the hydrogel increased less when the iron ion concentration increased to 0.04 M. However, the hysteresis curve increased at 60% compressive strain. A “fault” was visible in the cyclic compression curve (Figure 3h). This demonstrated that the hydrogel had difficulty returning to its initial form in time, and the energy dissipation was not timely at 0.04 M Fe^3+^, which was macroscopically manifested as “hard and brittle”. The inhomogeneity of the prepolymerized solution increased further with the concentration of Fe^3+^. Its colloidal properties were affected due to excessive viscosity. The swelling performance of hydrogels with different iron concentrations was explored. The swelling ratio was the largest at 0.03 M Fe^3+^. Therefore, the optimal Fe^3+^ concentration is 0.03 M. In addition, the compressive strain recovery performance of HC-CSN-Fe^3+^ hydrogel under optimal conditions under external force was explored (Figure 3i). It was found that the hydrogel had good compressive resistance and recovery performance [42].

### 2.7. Analysis of Water Retention Properties of HC-CSN-Fe^3+^ Hydrogels

Figure 4a displays the water retention capacity of different hydrogels in the air. HC-CSN-Fe^3+^ hydrogels maintain the best water retention capacity within 12–48 h compared with HC-PAA hydrogels and HC-CSN hydrogels. The water retention ratio was 66.84% at 12 h. It was 2.85% at 48 h. This was attributed to a denser and richer network that confers less water loss [43]. Figure 4b exhibits the water retention capacity of HC-CSN-Fe^3+^ hydrogel in soil. It was 97.54% at 48 h. The addition of CSN and Fe^3+^ can improve the water retention performance of hydrogels, which can be used to improve the water utilization and drought resistance of plants [7]. In addition, good biocompatibility and environmental friendliness provide hydrogels with greener application potential in soil [44,45,46].

### 2.8. Rheological Analysis of HC-CSN-Fe^3+^ Hydrogels

The crosslinking mode of hydrogel is an important factor affecting its rheological properties. Therefore, the rheological properties of HC-CSN-Fe^3+^ hydrogel were tested, as shown in Figure 1. Figure 1d shows that the storage modulus G’ was always higher than the loss modulus G” in the frequency range of 0.1–10 Hz. This indicates that the hydrogel was in a stable gel state with elastic behavior. As shown in Figure 1e, the viscosity of HC-CSN-Fe^3+^ hydrogel decreased sharply with the shear rate. This was attributed to the disruption of fragile cluster structures inside the hydrogel and reflects the shear-thinning behavior of the hydrogel.

### 2.9. Analysis of Antioxidant Properties of HC-CSN-Fe^3+^ Hydrogels

The antioxidant properties of the hydrogels were evaluated by measuring the inhibition ratio of free radicals, as shown in Figure 1g. The inhibition ratios of different hydrogels against 1, 1-Diphenyl -2-picrylhydrazyl radical (DPPH) were 46.99%, 66.12%, and 76.75% for HC-CSN hydrogel, HC-CSN-Fe^3+^ hydrogel, and HC-CSN-Fe^3+^ hydrogel, respectively, after 1 h. The improvement in the free radical inhibition ratio of HC-CSN and HC-CSN-Fe^3+^ hydrogels can be attributed to the antioxidant properties of CSN. In addition, HC-CSN-Fe^3+^ hydrogels showed a high DPPH radical inhibition ratio. This was attributed to the presence of metal ions, which also enhanced the antioxidant properties of the samples. The antioxidant properties contribute to the long-term application and preservation potential of the hydrogel without being compromised.

### 2.10. Analysis of UV Resistance of HC-CSN-Fe^3+^ Hydrogels

Conventional hydrogels are often limited in their range of applications due to their lack of UV shielding, especially in extreme environments. The strongest radiation wavelength of UV light is in the range of 320–400 nm, while the wavelength most sensitive to human eyes in the visible light range is 550 nm. Therefore, the absorbance values at 356 nm and 550 nm were chosen to calculate the UV shielding efficiency and transmittance of the hydrogels, as shown in Figure 1f. In the UV region at 365 nm, the UV transmittance of different hydrogels were 86.7%, 38.9%, and 7.8%, respectively. This shows that the introduction of CSN can effectively improve the UV shielding effect. In particular, the HC-CSN-Fe^3+^ hydrogel had the best UV shielding performance. This is due to the presence of Fe^3+^. In the visible light region, the transmittance of visible light was 93.1%, 91.3%, and 40.2% for different hydrogels, respectively. HC-CSN-Fe^3+^ hydrogel has a transmittance of 40.2% in the visible range. The results show that the hydrogel can effectively shield UV while maintaining excellent transparency. The application value of hydrogel in extreme environments is enhanced by enhancing its UV shielding properties, especially in areas with high UV radiation.

### 2.11. Analysis of Urea Release Performance of HC-CSN-Fe^3+^/SA Core-Shell Hydrogels

The cumulative release ratio of urea from hydrogel spheres coated with SA and uncoated SA in water is shown in Figure 4. The cumulative release ratio of urea from uncoated hydrogel spheres was 31.16% (1 h), 47.06% (4 h), 58.40% (8 h), and 65.29% (12 h), respectively. In contrast, the cumulative release ratio of urea from SA-coated hydrogel spheres was 18.60% (1 h), 25.14% (4 h), 26.62% (8 h), and 27.42% (12 h), respectively. Urea was released from SA-coated hydrogel spheres at a higher ratio within 0–3 h. This is due to the rapid release of urea from the surface of the hydrogel sphere through the first SA membrane. Additionally, there is a greater concentration differential between the hydrogel sphere’s interior and exterior, which causes a faster release. The urea release ratio leveled off over time. This was attributed to the gradual reduction of the concentration difference [47]. In fact, releasing urea from the inside of the hydrogel sphere requires overcoming the electrostatic interaction between SA and CSN. The release ratio of the hydrogel was greatly reduced, and urea was released more slowly, effectively solving the problem of urea waste.

### 2.12. Analysis of Urea Release Kinetics by HC-CSN-Fe^3+^/SA Core-Shell Hydrogel in Water

Figure 5 shows the fitting curves of different kinetic models of urea release by hydrogel pellets coated and uncoated with SA in water. The fitting results are shown in Table 2.

The optimal kinetic model for urea release from water-based hydrogel spheres, with and without SA encapsulation, was determined by fitting curves and calculating correlation coefficients. The Korsmeyer–Peppas model, which models drug release kinetics based on changes in diffusion coefficients of water and drug, was used to study the drug release dynamics of the composite hydrogel. The results showed that drug release was controlled by Fick diffusion. The Higuchi model is a theoretical model for the release properties of water-soluble drugs in semi-solid/solid preparations. It is widely used to study the dissolution ratio of sustained-release agents. The model assumes that the diffusion coefficient of the drug is constant. It is, therefore, suitable for diffusion through many curved pores, but not for preparations with significant changes in shape or surface coating. The correlation coefficients of the fitting curves for urea release from the hydrogel spheres without SA encapsulation were 0.9939, 0.9900, 0.9146, and 0.9523. These findings exhibited that the non-SA encapsulated hydrogel pellets’ drug release kinetics were better in line with the Korsmeyer–Peppas model, and the diffusion index (n) was 0.3098. This is consistent with Fickian diffusion when n < 0.45 [7]. The correlation coefficients of the fitting curves for urea release from the hydrogel spheres with SA encapsulation were 0.9015, 0.9547, 0.8403, and 0.8477. It was found that the Higuchi model had the highest correlation coefficient. This indicates that urea release in SA-coated hydrogel pellets no longer followed the Korsmeyer–Peppas model. The urea release mechanism after SA encapsulation was changed. This is because the release of urea from the SA-coated hydrogel sphere requires not only the release of urea from the hydrogel sphere but also overcoming the barrier formed by the outer layer of SA and CSN due to electrostatic interaction. In addition, compared to the hydrogel spheres without SA encapsulation, the hydrogel spheres with SA encapsulation had a smaller diffusion ratio constant (0.0973). This means that the slow-release ratio of urea is controlled to achieve a controlled release of urea.

### 2.13. Analysis of Urea Release Kinetics by HC-CSN-Fe^3+^/SA Core-Shell Hydrogels in Soil

Figure 6 shows the fitting curves of different kinetic models of urea release from hydrogel pellets coated and uncoated with SA in soil. The fitting results are shown in Table 3. The best kinetic model of urea release from the soil by the correlation coefficient of the fitted curve was used to determine the optimal kinetic model of urea release from the soil by hydrogel pellets coated and uncoated with SA.

The fitting results (Table 3) showed that the correlation coefficients of the release kinetics fitting curves of non-encapsulated sodium alginate hydrogel beads for urea were 0.9690, 0.9958, 0.9385, and 0.9528. The results showed that urea release kinetics from unencapsulated SA hydrogel beads was consistent with the Higuchi model. On the other hand, the correlation coefficients of the release kinetics fitting curves of SA-encapsulated hydrogel beads for urea were 0.8794, 0.9894, 0.9563, and 0.9595. These results indicated that the Higuchi model had the highest correlation coefficient among the four kinetic models. In addition, SA-encapsulated hydrogel beads obtained a smaller diffusion ratio constant (0.0288) compared to non-encapsulated SA hydrogel beads. The results showed that sustained urea release was achieved. This work also encouraged us to explain the potential computational simulation study of biomaterials investigation. It brings several advantages compared to experimental investigation, such as lower cost and faster results. This computational simulation would also become a preliminary study [48,49,50,51,52,53].

## 3. Materials and Methods

### 3.1. Materials

HC, ferric chloride hexahydrate, AA, CSN, ammonium persulfate, potassium bromide, N,N’ -methylenebisacrylamide (MBA), ammonium persulfate (APS), sodium alginate, DPPH, anhydrous ethanol, and p-dimethylaminobenzaldehyde (PDAB) were purchased from Aladdin Biotechnology Co., Ltd. (Shanghai, China).

### 3.2. Preparation of Hemicellulose-Based Hydrogels and Preparation of Sustained-Release Urea Systems

First, 1.0 g HC, 0.8 g CSN, and 40 mL of ferric chloride solution (0.03 M) were added in a 250 mL three-mouth flask and stirred for 20 min at 60 °C. Next, the liquid was cooled to room temperature and stirred continuously for 5 min while AA and APS were added under a nitrogen atmosphere. Finally, 0.04 g MBA was added and stirred until completely dissolved to obtain the prepolymer solution. The crosslink was dried at 70 °C for 2 h to obtain hemicellulose–chitosan–iron ion hydrogel (HC-CSN-Fe^3+^). HC-PAA and HC-CSN hydrogels were prepared in the same step.

Preparation of urea sustained-release hydrogel: firstly, urea was loaded into the hydrogel system. The prepared hydrogel ball was soaked in sodium alginate solution and prepared by electrostatic interaction connection. The unreacted drug on the surface of the hydrogel pellets was washed off and the pellets soaked in sodium alginate solution at 1.5% wt for 6 h. The hydrogel pellets were freeze-dried for 24 h to obtain HC-CSN-Fe^3+^/SA core–shell hydrogel.

### 3.3. Characterization Analysis of HC-CSN-Fe^3+^ Hydrogels

The surface morphology and void characteristics of the hydrogel were observed under an SEM (Hitachi SU8220, Hitachi Limited, Tokyo, Japan). FT-IR (TENSOR II, Bruker, Karlsruhe, Germany) was used to determine the main functional group changes of different hydrogels in the measurement band range of 4000–500 cm^−1^. The chemical elemental composition and bond state of the sample was analyzed by XPS (ESCALAB 250XI+, Thermo Fisher Scientific, MA, USA).

### 3.4. Analysis of Compression Properties of HC-CSN-Fe^3+^ Hydrogels

The compressive strain resistance of cubic hydrogels (15 mm × 15 mm × 15 mm) was determined using an electron universal material testing machine (Instron bluehill LE, Instron, Boston, MA, USA). The cubic hydrogels were formed in a mold (15 mm × 15 mm × 15 mm) by the polymerization of polymer solutions. The compression strain ranged from 0 to 60% and the compression rate was constant at 5 mm·min^−1^. Three samples were taken in each group as parallel samples.

### 3.5. Analysis of Swelling Ratio of HC-CSN-Fe^3+^ Hydrogels

Small pieces of hydrogel samples were accurately weighed to a certain mass and recorded as W_d_. They were placed in ultrapure water at room temperature. Samples were taken at intervals of 8 h and wiped to remove free water from the surface until the sample mass was constant. This was weighed and recorded as W_t_ in three parallel samples per group. The swelling ratio (SR) of the hydrogel was calculated using Equation (1).
(1)$$\mathrm{SR}\left(\%\right)=\frac{W_{t}\;{-\;W}_{d}}{W_{d}}\;\times \;100\%$$

### 3.6. Analysis of Water Retention Properties of HC-CSN-Fe^3+^ Hydrogels

A total of 1 g of absolute dry hydrogel sample was dissolved with deionized water to swell equilibrium. The total water absorption was determined and placed in an empty beaker. The residual mass of the hydrogel was measured every 12 h at room temperature. The water retention performance W_r_ (%) was calculated according to Equation (2):(2)$$W_{r}=\frac{W_{0}{\;-\;W}_{t}}{W_{0}\;-\;W}\;\times \;100\%$$
where W_r_ is the water retention rate, W_0_ is the mass of the hydrogel during swelling equilibrium, W_t_ is the remaining mass of the hydrogel after different times, and W is the absolute dry mass of the hydrogel.

### 3.7. Rheological Analysis of HC-CSN-Fe^3+^ Hydrogels

The dynamic rheological properties of hydrogels were analyzed using a modular rheometer (HAAKE MARS4, Thermo Fisher Scientific, MA, USA). The hydrogel adhesion was analyzed as a function of the shear ratio. Strain amplitude was (γ = 0.1–1000%) and dynamic frequency sweep range was 0.1–10 Hz.

### 3.8. Analysis of Antioxidant Properties of HC-CSN-Fe^3+^ Hydrogels

The DPPH radical inhibition ratio was used as the analysis index of the antioxidant performance of the sample. The DPPH ethanol solution with concentration gradients of 0, 0.05, 0.10, 0.015, 0.02, and 0.025 g·L^−1^ was prepared. The absorbance value of different concentrations of DPPH ethanol solution at 517 nm was measured. The standard curve was drawn. Next, 0.1 g of powder sample and 10 mL of absolute ethanol were mixed for 3 h at room temperature in a 20 mL brown vial. Then, 2 mL of solution was mixed with 2 mL of 0.025 g·L^−1^ DPPH ethanol solution. It was placed in a dark place for 1 h to determine the absorbance of different samples at 517 nm. The DPPH radical suppression ratio was calculated according to Equation (3):(3)$$I\left(\%\right)=\frac{C_{1}\;{-\;C}_{2}}{C_{1}}\times \;100\%$$
where I is the DPPH radical inhibition ratio, C_1_ is the concentration of DPPH after the reaction, and C_2_ is the concentration of DPPH before the reaction.

### 3.9. Analysis of UV Resistance of HC-CSN-Fe^3+^ Hydrogels

Different hydrogel samples (2 cm × 1 cm × 2 mm) were placed in an ultraviolet spectrophotometer (Cary 3500, Agilent, Palo Alto, USA). The absorbance at the 300–800 nm range was measured. UV resistance was calculated according to Equation (4):(4)$$T_{t}\left(\%\right)=\frac{T_{2}}{T_{1}}\;\times \;100\%$$
where T_t_ is the UV transmittance, T_2_ is the total transmitted luminous flux through the sample, and T_1_ is the incident luminous flux.

### 3.10. Analysis of Urea Release Performance of HC-CSN-Fe^3+^/SA Core-Shell Hydrogels

First, a standard solution of urea was prepared and a standard solution curve was obtained. A total of 0.1 g of urea was dissolved in 100 mL of deionized water and stored at 4 °C. Next, 50 g·L^−1^ PDAB ethanol solution and 2 mol·L^−1^ sulfuric acid were prepared. The urea content in the solution was determined by the color reaction of PDAB with urea under acidic conditions. The standard curve was fitted. Then, 10 mL of PDAB ethanol solution and 4 mL of sulfuric acid were added to the urea standard solution. The volume was fixed with deionized water. The absorbance at 422 nm was measured using deionized water as a contrast sample. The urea content in the solution after different soaking time was determined by hydrostatic soaking method. A total of 1.0 g urea was added to 40 mL polymer solution in advance. The copolymerization was carried out to obtain urea-loaded hydrogel after mixing. The dried sample of the urea-loaded hydrogel sphere was placed in a beaker and 40 mL of deionized water was added. Next, 1 mL of solution was dropped into a 25 mL volumetric flask every 1 h, and then 10 mL of PDAB ethanol solution and 4 mL of sulfuric acid were added. The volume of deionized water was fixed. The cumulative release of urea in the aqueous solution was calculated. The sample of the hydrogel ball loaded with urea was placed in a 250 mL beaker, and 100 mL of deionized water and 50 g of dry sand were added after drying. The cumulative release ratio of urea was measured in the same way as above.

### 3.11. Release Kinetics Analysis of Urea using HC-CSN-Fe^3+^/SA Core-Shell Hydrogels

Generally, the release models of semi-solid preparations include the Korsmeyer–Peppas model, Higuchi model, zero-order model, and first-order model, and the mediator type of semi-solid preparation and drug delivery model are different [30]. The Korsmeyer–Peppas model, Higuchi model, zero-order model, and first-order model were used to model fertilizer release kinetics.


(1)Korsmeyer–Peppas model


(5)$$\frac{M_{t}}{M}{=\mathrm{kt}}^{n}$$
where M_t_/M is the release ratio of urea, k is the diffusion ratio constant, t is the release time, and n is the diffusion index.


(2)Higuchi model


(6)$$\frac{M_{t}}{M}{=k}_{h}t^{0.5}\;$$
where M_t_/M is the release ratio of urea, k_h_ is the diffusion ratio constant, and t is the release time.


(3)Zero-order model


(7)$$\frac{M_{t}}{M}{\;=k}_{0}t$$
where M_t_/M is the release ratio of urea, k_0_ is the diffusion ratio constant, and t is the release time.

(4)First-order model

(8)$$\frac{M_{t}}{M}{\;=\;1\;-\;e}^{{-k}_{1}t}$$
where M_t_/M is the release ratio of urea, k_1_ is the diffusion ratio constant, and t is the release time.

Release kinetic parameters were obtained by plotting ln(M_t_/M) and ln(t), M_t_/M and t^0^.^5^, M_t_/M and t, and −ln (1 − M_t_/M) and t images.

## 4. Conclusions

In this work, the water retention and sustained release characteristics of an HC-based hydrogel were prepared. The compression performance of the hydrogel was 350% better than the traditional HC-based hydrogel under the conditions of HC/CSN 5:3, 4 mL AA addition, and 0.03 M iron ion concentration. Furthermore, the hydrogel exhibited high water retention (97.54%) in the soil after 48 h. In addition, the antioxidant performance (76.75%) and UV resistance (92.2% UV blocking efficiency) of the hydrogel were enhanced. The release ratio of urea in water and soil decreased after coating the hydrogel with SA. Simulation of urea release kinetics in soil showed that the release kinetics of HC-CSN-Fe^3+^/SA core–shell hydrogel better fit the Higuchi model, with smaller kinetic constants. The results indicated that hydrogel had a significant slow-release effect on urea.

## Figures and Tables

**Figure 1 ijms-24-09208-f001:**
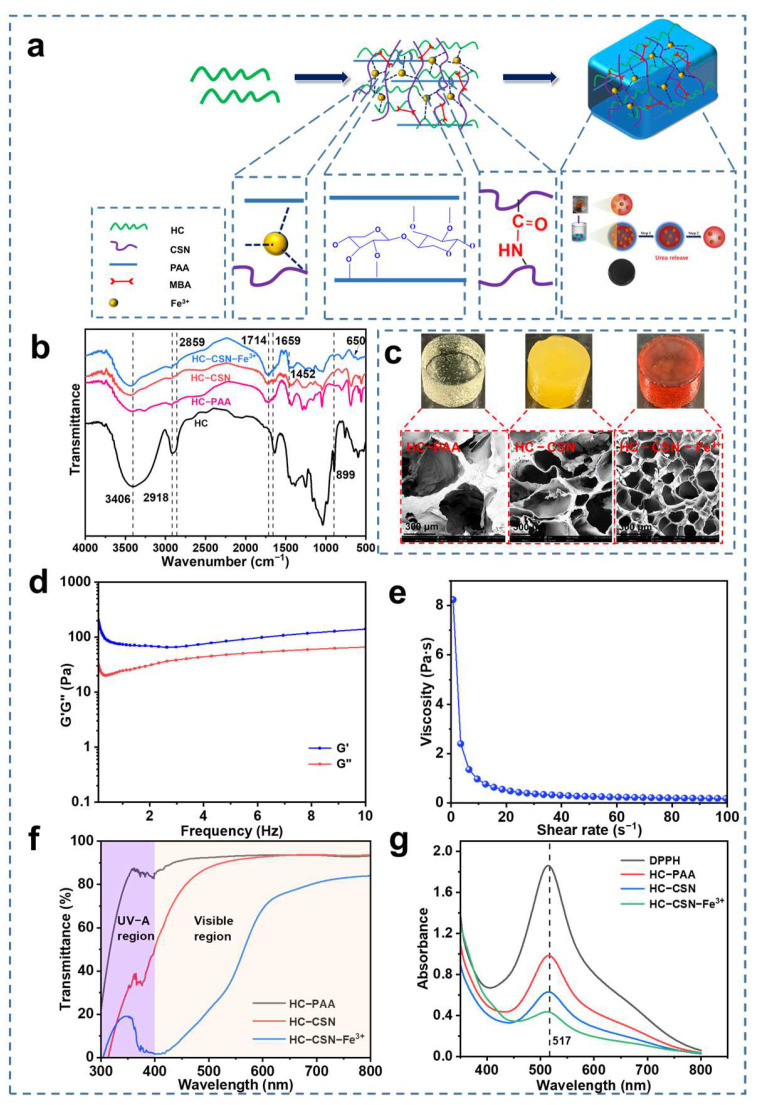
Preparation of HC-CSN-Fe^3+^ hydrogel ((**a**), FT-IR; (**b**), SEM; (**c**), rheological properties of HC-CSN-Fe^3+^ hydrogels; (**d**), modulus; (**e**), viscosity; (**f**), UV-shielding properties of different hydrogels; (**g**), DPPH radical inhibition ratio of different hydrogels).

**Figure 2 ijms-24-09208-f002:**
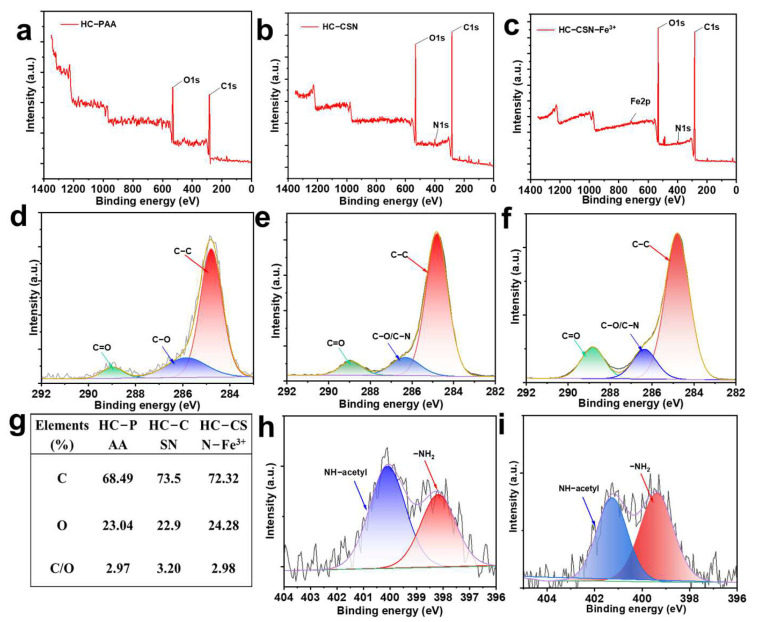
XRD of different hydrogels ((**a**), HC-PAA hydrogels; (**b**,**c**), HC-CSN hydrogels and HC-CSN-Fe^3+^ hydrogels; (**d**), C1s peak of HC-PAA hydrogel; (**e**,**h**), C1s and N1s peaks of HC-CSN hydrogel; (**f**,**i**), HC-CSN-Fe^3+^ hydrogel C1s peak and N1s peak); (**g**), The element content of different hydrogels.

**Figure 3 ijms-24-09208-f003:**
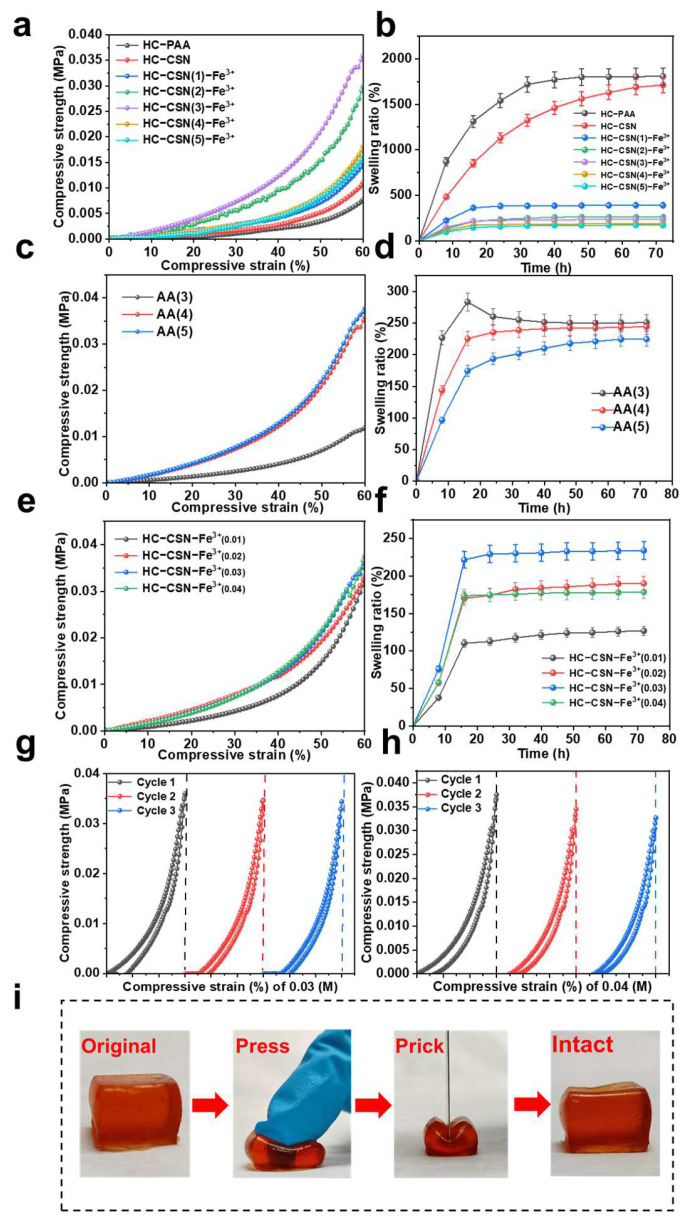
Preparation and property analysis of hydrogels ((**a**,**c**,**e**); effects of different component addition amounts on swelling property of hydrogels; (**b**,**d**,**f**), effects of different component addition levels on the compression properties of hydrogels; (**g**,**h**), compression cycle curve; (**i**), deformation recovery performance of hydrogel under compression).

**Figure 4 ijms-24-09208-f004:**
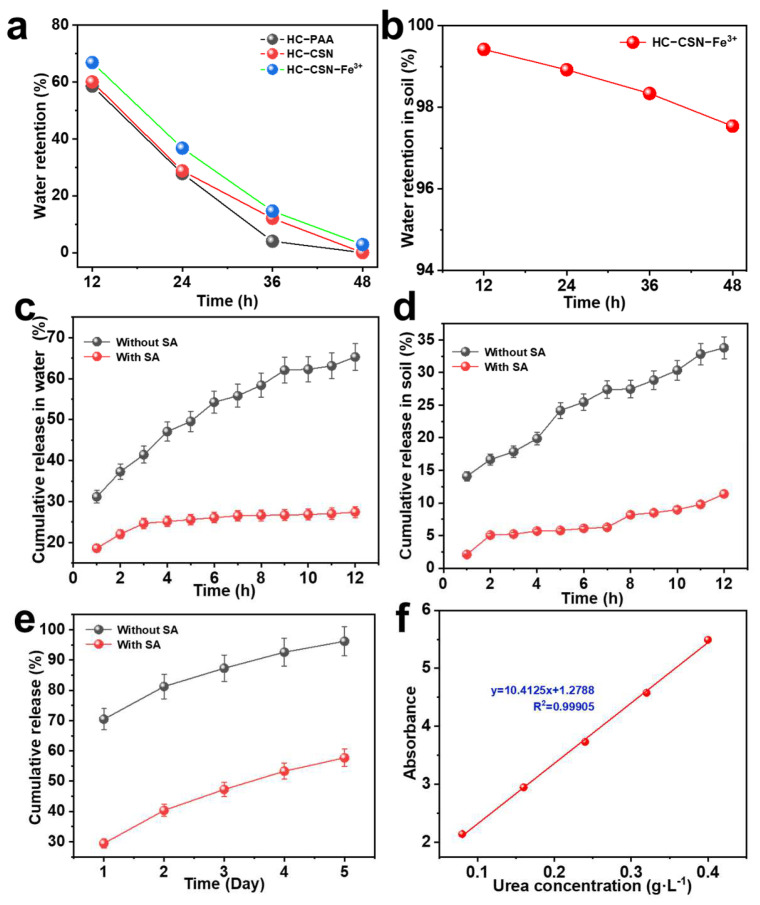
The water retention capacity of different hydrogels ((**a**), in air; (**b**), in soil; (**c**,**e**), the cumulative release ratio of urea from hydrogel in water; (**d**), the cumulative release ratio of hydrogel to urea in soil; (**f**), standard curve).

**Figure 5 ijms-24-09208-f005:**
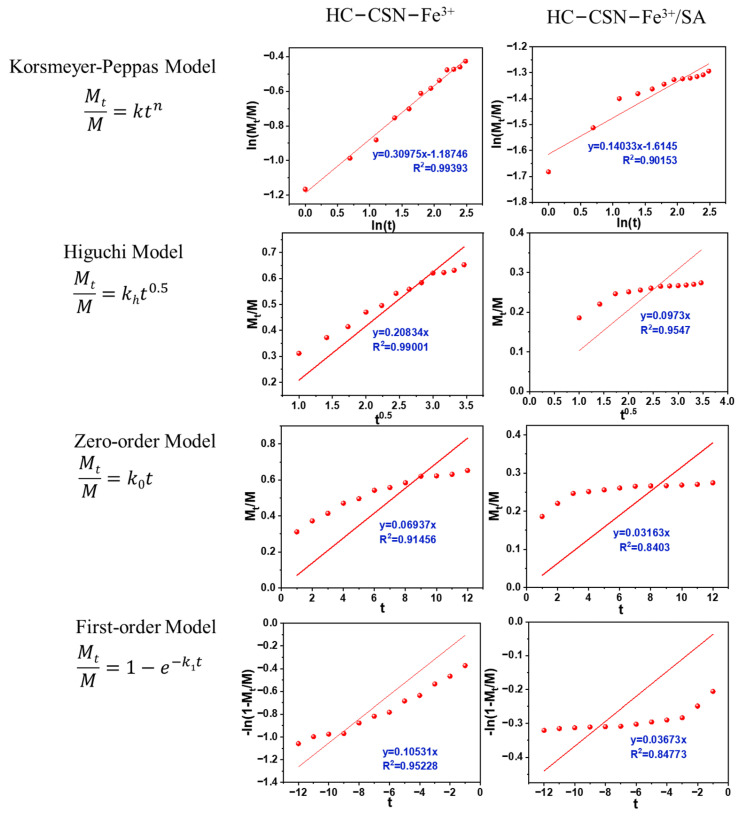
Model fitting of urea release kinetics in water of different hydrogels.

**Figure 6 ijms-24-09208-f006:**
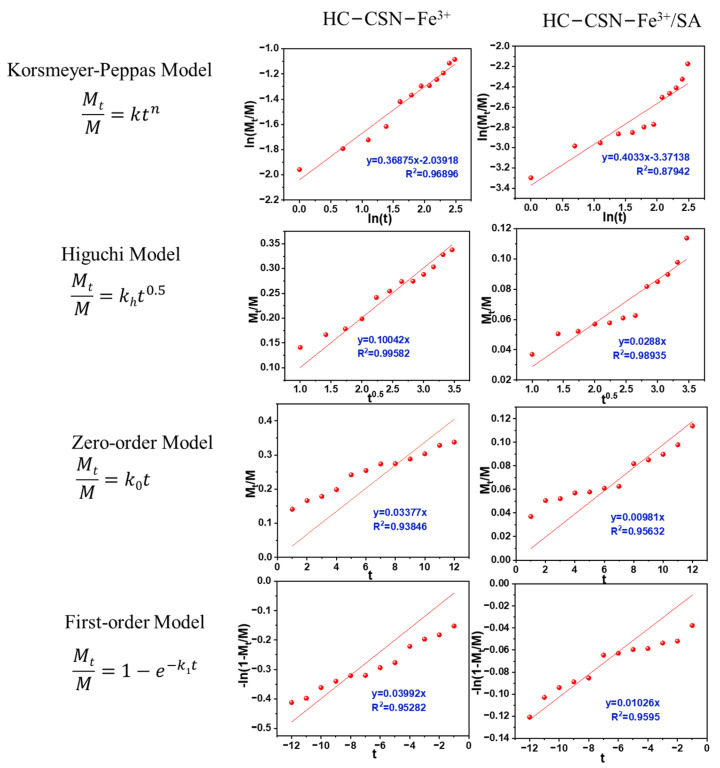
Model fitting of urea release kinetics in soil of different hydrogel.

**Table 1 ijms-24-09208-t001:** Effect of different content components on hydrogels.

Samples	HC/CSN Ratio (g·.g^−1^)	Fe^3+^ (mol·L^−1^)	AA (mL)
HC-CSN-Fe^3+^	5:1	0.03	4
HC-CSN-Fe^3+^	5:2	0.03	4
HC-CSN-Fe^3+^	5:3	0.03	4
HC-CSN-Fe^3+^	5:4	0.03	4
HC-CSN-Fe^3+^	5:5	0.03	4
HC-CSN-Fe^3+^	5:3	0.03	3
HC-CSN-Fe^3+^	5:3	0.03	4
HC-CSN-Fe^3+^	5:3	0.03	5
HC-CSN-Fe^3+^	5:3	0.01	4
HC-CSN-Fe^3+^	5:3	0.02	4
HC-CSN-Fe^3+^	5:3	0.03	4
HC-CSN-Fe^3+^	5:3	0.04	4

**Table 2 ijms-24-09208-t002:** Fitting parameters of different kinetic models in water.

Kinetic Models	Parameter	Gel	Gel-SA
Korsmeyer-Peppas	R^2^	0.9939	0.9015
	n	0.3098	0.1403
	k	0.3050	0.1990
Higuchi	R^2^	0.9900	0.9547
	k_h_	0.2083	0.0973
Zero-order	R^2^	0.9146	0.8403
	k_0_	0.0694	0.0316
First-order	R^2^	0.9523	0.8477
	k_1_	0.1053	0.0367

**Table 3 ijms-24-09208-t003:** Fitting parameters of different kinetic models in soil.

Kinetic Models	Parameter	Gel	Gel-SA
Korsmeyer-Peppas	R^2^	0.9690	0.8794
	n	0.3688	0.4033
	k	0.1301	0.0343
Higuchi	R^2^	0.9958	0.9894
	k_h_	0.1004	0.0288
Zero-order	R^2^	0.9385	0.9563
	k_0_	0.0338	0.0098
First-order	R^2^	0.9528	0.9595
	k_1_	0.3992	0.0103

## Data Availability

The data presented in this study are available in the manuscript’s figure.

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
