# Peer review of "High-Strength, High-Water-Retention Hemicellulose-Based Hydrogel and Its Application in Urea Slow Release"

_ijms, 2023, doi:10.3390/ijms24119208_

Round 1
Reviewer 1 Report
The manuscript by Hou et al. is devoted to synthesis of hemicellulose-chitosan based reinforced by ferric (III) ions. The authors research how the presence of iron ions affects mechanical properties of the SAPs as well as utilize the SAPs obtained as urea-controlled release device by the coating by sodium alginate.
The suggested approach is multi-stage, this can complicate the practical use of the SAPs obtained.
lines 112, 1659 cm-1 is C=O of chitosan acetyl groups stretching, not O-H vibrations.
At some figures the authors used abbreviation Xyl, however, it didn’t descript in the text. Also, what is PDAB and DPPH?
Fig. 5 and 6 have poor resolution and unreadable.
Lines 246-254 and 255-263 contain the same text.
The manuscript can be accepted after above-mention correctios.
Extensive editing of English language required
Author Response
Dear Reviewer,
Thank you for your letter and for the comments concerning our manuscript entitled “High strength, high water retention hemicellulose based hydrogel and its application in urea slow release”. We have studied your comments carefully and have made corrections which we hope could meet your requirements. All changes were marked up using the “Track Changes” function.
Questions you put forward are explained as follows:
- The suggested approach is multi-stage, this can complicate the practical use of the SAPs obtained.
This work selected several natural polymer polysaccharides as the main material, which are degradable and biocompatible. Different components give hydrogels different properties. Chitosan, as a natural polysaccharide with excellent antioxidant and ultraviolet resistance, can improve the mechanical properties and water retention of traditional hemicellulose hydrogels through the connection of chemical bonds and ionic bonds. As a polyanionic natural polysaccharide, sodium alginate can form stable structure with polycationic polysaccharide chitosan through ionic bond to slow down the release rate of urea. Although the process is multi-stage, the preparation and connection of each stage is relatively simple and convenient.
- Lines 112, 1659 cm-1 is C=O of chitosan acetyl groups stretching, not O-H vibrations.
It was modified according to the comments.
- At some figures the authors used abbreviation Xyl, however, it didn’t descript in the text. Also, what is PDAB and DPPH?
Xyl has been replaced by HC(Hemicellulose) and the figures have been modified. PDAB stands for p-dimethylaminobenzaldehyde, which can undergo color reaction with urea under acidic conditions and can be used to determine urea content. DPPH stands for 1, 1-DiPhenyl -2-picrylhydrazyl radical, used to test the antioxidant capacity of sample. These have been explained in the Materials and Methods.
- Fig. 5 and 6 have poor resolution and unreadable.
It was modified according to the comments.
- Lines 246-254 and 255-263 contain the same text.
It was deleted according to the comments.
- Extensive editing of English language required
The language in the manuscript was elevated. The language editing services are provided by Elsevier. Order reference: ASLESTD0433068
As a whole, issues the referee suggested are very pertinent, which are very helpful to modify my entire paper and thank you very much again.
Reviewer 2 Report
In this work, acrylic monomer was grafted onto the hemicellulose molecular chain 78 by free radicals, introduced into chitosan-ferric chloride system, and a novel bio-based hydrogel (HC-CSN-Fe3+) with envelope structure was prepared by coating sodium alginate for sustained urea release. I am giving several comments to the authors in this round.
1. Line 29, As arable land continues to decrease and the world's population grows, please give/mention quantitative data.
2. Line 34 stated that “countries and regions”, please give the most of them?
3. Line 51 related in the Introdcution section, some general information for biodegradable concept and explanation should be provided. It would make it clearly to the reader improving its understood. Also, suggested reference needs to adopted as follows, doi: 10.3390/biomedicines11020427 and 10.3390/su15010823
4. Line 55-57 related In the Introduction section, the authors encouraging to explain in brief about biocompability of biomaterials. It is one of the great features in materials. Also, linked explanation in discussion section also needed. Please provide this information along with relevant reference as follows, doi: 10.3390/biomedicines11030951, 10.3390/su142013413, and 10.3390/ma14247554
5. Line 408-411, where is the basis of this equation Analysis of water retention properties of HC-CSN-Fe3+ hydrogels adopted? Please give it and make a rationalitation of explanation.
6. The novelty in the current article by the authors is too weak. The past has seen extensive published work of written material. It is required to provide more details for more explanation about the present novel in the introductory section.
7. The work, novelty, and constraints of relevant previous literature must be explained in the introduction section to highlight the article gaps that the present work aims to fill.
-
Author Response
Dear Reviewer,
Thank you for your letter and for the comments concerning our manuscript entitled “High strength, high water retention hemicellulose based hydrogel and its application in urea slow release”. We have studied your comments carefully and have made corrections which we hope could meet your requirements. All changes were marked up using the “Track Changes” function.
Questions you put forward are explained as follows:
- Line 29, as arable land continues to decrease and the world's population grows, please give/mention quantitative data.
More detailed explanation of the data has been supplemented.
The world's population increased from 6.1 billion to 7.8 billion between 2000 and 2020. However, the arable land decreased by 75 million hectares according to the 2020 Statistical Year-book of World Food and Agriculture.
- Line 34 stated that “countries and regions”, please give the most of them?
More detailed explanation of the data has been supplemented.
India is the world's largest importer of urea in 2020, accounting for 20.4%.
- Line 51 related in the Introdcution section, some general information for biodegradable concept and explanation should be provided. It would make it clearly to the reader improving its understood. Also, suggested reference needs to adopted as follows, doi: 10.3390/biomedicines11020427 and 10.3390/su15010823
The biodegradable concept and explanation have been supplemented and the suggested references have been adopted.
- Line 55-57 related In the Introduction section, the authors encouraging to explain in brief about biocompability of biomaterials. It is one of the great features in materials. Also, linked explanation in discussion section also needed. Please provide this information along with relevant reference as follows, doi: 10.3390/biomedicines11030951, 10.3390/su142013413, and 10.3390/ma14247554
The biocompability concept and explanation have been supplemented in discussion section and the suggested references have been adopted.
- Line 408-411, where is the basis of this equation Analysis of water retention properties of HC-CSN-Fe3+ hydrogels adopted? Please give it and make a rationalitation of explanation.
Thanks to the reviewer for the question. The water retention properties can be calculated by the equation 3-2. The detailed steps can be seen in reference [7,14].
- The novelty in the current article by the authors is too weak. The past has seen extensive published work of written material. It is required to provide more details for more explanation about the present novel in the introductory section.
More details for more explanation about the present novel in the introductory section have been added.
- The work, novelty, and constraints of relevant previous literature must be explained in the introduction section to highlight the article gaps that the present work aims to fill.
This work fills a gap in the application of hemicellulose for urea slow-release materials. Hemicellulose-based hydrogels have great application potential in the field of urea sustained release and are expected to be used as a good biomass-based slow-release fertilizer. More details have been added in the introduction section.
As a whole, issues the referee suggested are very pertinent, which are very helpful to modify my entire paper and thank you very much again.
Reviewer 3 Report
In this manuscript, the authors reported properties of hemicellulose based hydrogels which could be used as an effective fertilizer with slow urea release properties. Combination of several methods (FT-IR spectroscopy, SEM, XPS) is used for hydrogels characterization.
The methodology is described correctly. The results are clear and well explained. However, the article needs minor changes to be published in IJMS.
Some remarks:
Line 93: abbreviations should be explained
Line 139: table in Fig 2g is unreadable (too small font).
Line 203: should be ”hemicellulose”.
Lines 246-254: the same text repeated in lines 255-263
Line 294: abbreviations should be explained
Line 316 and 347: fitted equations in Fig.5 and Fig. 6 are totally unreadable
Line 382: I think that it should be explained how urea was loaded into the hydrogel system
Line 397: Please explain how cubes for compression tests were obtained
Line 398: I don’t understand what means “electron universal testing machine”. Electrons are used in compression testing?
Line 401: The chapter title is not relevant to the content
Line 441: abbreviations should be explained
Author Response
Dear Reviewer,
Thank you for your letter and for the comments concerning our manuscript entitled “High strength, high water retention hemicellulose based hydrogel and its application in urea slow release”. We have studied your comments carefully and have made corrections which we hope could meet your requirements. All changes were marked up using the “Track Changes” function.
Questions you put forward are explained as follows:
- Line 93: abbreviations should be explained
It was modified according to the comments. Full names of all abbreviations are provided when they first appear.
- Line 139: table in Fig. 2g is unreadable (too small font).
It was modified according to the comments.
- Line 203: should be “hemicellulose”.
It was modified according to the comments.
- Lines 246-254: the same text repeated in lines 255-263
It was deleted according to the comments.
- Line 294: abbreviations should be explained
It was modified according to the comments. Full names of all abbreviations are provided when they first appear.
- Line 316 and 347: fitted equations in Fig.5 and Fig. 6 are totally unreadable
It was modified according to the comments.
- Line 382: I think that it should be explained how urea was loaded into the hydrogel system
It was modified according to the comments. For quantitative analysis of urea before and after release, 1.0g urea was added to 40mL polymer solution in advance. Copolymerization was carried out to obtain urea-loaded hydrogel after mixing well.
- Line 397: Please explain how cubes for compression tests were obtained
The cubic hydrogels were obtained in a mold (15 mm*15 mm*15 mm) by the polymerization of polymer solution.
- Line 398: I don’t understand what means “electron universal testing machine”. Electrons are used in compression testing?
“Electron” indicates “computer-controlled” Electron universal testing machine is the use of electronic technology (or computer) control of universal material testing machine. The instrument used in this work is completely controlled by the microcomputer, easy to operate, flexible, according to the display of the keyboard or mouse to implement the operation control of the testing machine.
- Line 401: The chapter title is not relevant to the content
The chapter title was modified to better match the content of the study.
- Line 441: abbreviations should be explained
It was modified according to the comments. Full names of all abbreviations are provided when they first appear.
As a whole, issues the referee suggested are very pertinent, which are very helpful to modify my entire paper and thank you very much again.
Round 2
Reviewer 2 Report
Following comments is given in the revised form for this manuscript.
1. Line 54, related to biomass utilization, incorporated related study of biomass as follows, doi: 10.12911/22998993/158564
2. Line 71, difference between AA and CSN not clearly understood.
3. Line 123, “248.8 eV, 285.8 eV, and 288.9 eV”, please recheck the values.
4. Line 130, “with addition of CSN”, Why giving this additional information? Not got the point for this changes.
5. Line 144, the explanation for Table 1 is quite simple, please make it more comprehensive.
6. The abstract section should include quantitative results.
7. Please conclude your abstract with a "take-home" message.
8. Keywords should be reordered based on alphabetical order.
9. The authors encouraged to explain regarding potential computational simulation study of biomaterials investigation. It brings several advantages compared to experimental investigation such as lower cost and faster results. Include the explanation along with supporting literature as follows, doi: 10.1016/j.heliyon.2022.e12050, 10.1038/s41598-023-30725-6, and 10.3390/met12081241
10. Extending the explanation, computational simulation also would become preliminary study before performing experimental study or supporting the results of experimental study. Incorporated relevant reference for the explanation as follows, doi: 10.3390/ma16093298, 10.1177/14657503221144810, 10.3390/fluids7070225
-
Author Response
Dear Reviewer,
Thank you for your letter and for the comments concerning our manuscript entitled “High strength, high water retention hemicellulose based hydrogel and its application in urea slow release”. We have studied your comments carefully and have made corrections which we hope could meet your requirements. All changes were marked up using the “Track Changes” function.
Questions you put forward are explained as follows:
- Line 54, related to biomass utilization, incorporated related study of biomass as follows, doi: 10.12911/22998993/158564
The suggested references have been adopted.
- Line 71, difference between AA and CSN not clearly understood.
Acrylic acid (AA) can be grafted onto hemicellulose chain by copolymerization. The amino group of CSN is dehydrated with the carboxyl group of acrylic acid to form an amide bond and improve the properties of hydrogels. The details can be seen in reference below.
Essawy, H.A.; Ghazy, M.B.M.; Abd El-Hai, F.; Mohamed, M.F. Superabsorbent hydrogels via graft polymerization of acrylic acid from chitosan-cellulose hybrid and their potential in controlled release of soil nutrients. Int. J. of Biol. Macromol. 2016, 89, 144-151, doi:10.1016/j.ijbiomac.2016.04.071
- Line 123, “248.8 eV, 285.8 eV, and 288.9 eV”, please recheck the values.
These values are drawn from the following reference: (Wang, H.X.; Qan, J.; Ding, F.Y. Emerging Chitosan-Based Films for Food Packaging Applications. J. Agric. Food Chem. 2018, 66, 395-413, doi:10.1021/acs.jafc.7b04528.)
- Line 130, “with addition of CSN”, Why giving this additional information? Not got the point for this changes.
The addition of CSN will be grafted onto the carbon chain through dehydration reaction, and the C-C peak will increase.
- Line 144, the explanation for Table 1 is quite simple, please make it more comprehensive.
This work designed a single factor experiment to explore the effects of different contents on hydrogels, as shown in Table 1. The HC/CSN ratio ranges from 5:1 to 5:5 (g.g-1), the concentration of Fe3+ ranges from 0.01 to 0.04 (mol. L-1) and the addition of AA ranges from 3 to 5 (mL). The information has been added.
- The abstract section should include quantitative results.
The abstract was revised in accordance with comments.
- Please conclude you’re abstract with a "take-home" message.
It was modified according to the comments.
- Keywords should be reordered based on alphabetical order.
It was modified according to the comments.
- The authors encouraged to explain regarding potential computational simulation study of biomaterials investigation. It brings several advantages compared to experimental investigation such as lower cost and faster results. Include the explanation along with supporting literature as follows, doi: 10.1016/j.heliyon.2022.e12050, 10.1038/s41598-023-30725-6, and 10.3390/met12081241
The suggested references have been adopted.
- Extending the explanation, computational simulation also would become preliminary study before performing experimental study or supporting the results of experimental study. Incorporated relevant reference for the explanation as follows, doi: 10.3390/ma16093298, 10.1177/14657503221144810, 10.3390/fluids7070225
The suggested references have been adopted.
As a whole, issues the referee suggested are very pertinent, which are very helpful to modify my entire paper and thank you very much again.
Round 3
Reviewer 2 Report
Well done to the auhtors for their effort.
-